# Effect of Organic Anion Transporting Polypeptide 1B1 on Plasma Concentration Dynamics of Clozapine in Patients with Treatment-Resistant Schizophrenia

**DOI:** 10.3390/ijms252313228

**Published:** 2024-12-09

**Authors:** Toshihiro Sato, Takeshi Kawabata, Masaki Kumondai, Nagomi Hayashi, Hiroshi Komatsu, Yuki Kikuchi, Go Onoguchi, Yu Sato, Kei Nanatani, Masahiro Hiratsuka, Masamitsu Maekawa, Hiroaki Yamaguchi, Takaaki Abe, Hiroaki Tomita, Nariyasu Mano

**Affiliations:** 1Department of Pharmaceutical Sciences, Tohoku University Hospital, Sendai 980-8574, Japan; masaki.kumondai.d5@tohoku.ac.jp (M.K.); yu.sato.e7@tohoku.ac.jp (Y.S.); masahiro.hiratsuka.a8@tohoku.ac.jp (M.H.); m-maekawa@tohoku.ac.jp (M.M.); nariyasu.mano.c8@tohoku.ac.jp (N.M.); 2Graduate School of Information Sciences, Tohoku University, Sendai 980-8578, Japan; takeshi.kawabata.b8@tohoku.ac.jp; 3Faculty of Pharmaceutical Sciences, Tohoku University, Sendai 980-8578, Japan; 4Department of Psychiatry, Tohoku University Hospital, Sendai 980-8574, Japan; hkomatsu1019@gmail.com (H.K.); goh.ong.1008@gmail.com (G.O.); htomita@med.tohoku.ac.jp (H.T.); 5Department of Psychiatry, Graduate School of Medicine, Tohoku University, Sendai 980-8575, Japan; ykikuchi@sand.ocn.ne.jp; 6Advanced Research Center for Innovations in Next-Generation Medicine, Tohoku University, Sendai 980-8573, Japan; kei.nanatani.a7@tohoku.ac.jp; 7Tohoku Medical Megabank Organization, Tohoku University, Sendai 980-8573, Japan; 8Department of Pharmacy, Yamagata University Hospital, Yamagata 990-9585, Japan; hiroaki.yamaguchi@med.id.yamagata-u.ac.jp; 9Graduate School of Medical Science, Yamagata University, Yamagata 990-9585, Japan; 10Division of Nephrology, Endocrinology, and Vascular Medicine, Graduate School of Medicine, Tohoku University, Sendai 980-8574, Japan; takaabe@med.tohoku.ac.jp; 11Division of Medical Science, Graduate School of Biomedical Engineering, Tohoku University, Sendai 980-8579, Japan; 12Department of Clinical Biology and Hormonal Regulation, Graduate School of Medicine, Tohoku University, Sendai 980-8575, Japan

**Keywords:** clozapine, norclozapine, treatment-resistant schizophrenia, OATP1B1, SNP, Japanese

## Abstract

The involvement of drug-metabolizing enzymes and transporters in plasma clozapine (CLZ) dynamics has not been well examined in Japanese patients with treatment-resistant schizophrenia (TRS). Therefore, this clinical study investigated the relationship between single nucleotide polymorphisms (SNPs) of various pharmacokinetic factors (drug-metabolizing enzymes and transporters) and dynamic changes in CLZ. Additionally, we aimed to determine whether CLZ acts as a substrate for pharmacokinetic factors using in vitro assays and molecular docking calculations. We found that 6 out of 10 patients with TRS and with multiple organic anion transporting polypeptide (OATP) variants (OATP1B1: **1b*, **15*; OATP1B3: 334T>G, 699G>A; and OATP2B1: **3*, 935G>A, 601G>A, 76_84del) seemed to be highly exposed to CLZ and/or *N*-desmethyl CLZ. A CLZ uptake study using OATP-expressing HEK293 cells showed that CLZ was a substrate of OATP1B1 with *K*_m_ and *V*_max_ values of 38.9 µM and 2752 pmol/mg protein/10 min, respectively. The results of molecular docking calculations supported the differences in CLZ uptake among OATP molecules and the weak inhibitory effect of cyclosporine A, which is a strong inhibitor of OATPs, on CLZ uptake via OATP1B1. This is the first study to show that CLZ is an OATP1B1 substrate and that the presence of SNPs in OATPs potentially alters CLZ pharmacokinetic parameters.

## 1. Introduction

In Japan, approximately 780,000 patients are afflicted with schizophrenia, and approximately 30% may have treatment-resistant schizophrenia (TRS) [1,2]. Clozapine (CLZ) is a medication used for TRS with a wide maintenance dose range of 200–600 mg and a substantial and well-known inter- and intra-individual variability in CLZ plasma concentration [3]. Insufficient efficacy occurs at low blood concentrations, whereas the risk of severe adverse events in the central nervous and cardiovascular systems increases at >1000 ng/mL [4]. The specified effective therapeutic range is 350–600 ng/mL (trough concentration) [4], and it has become eligible for a specific drug treatment management fee, a medical fee related to therapeutic drug monitoring, in the FY 2022 Revision of Medical Fees in Japan.

International guidelines recommend slow titration of CLZ in East Asians, including the Japanese, because their plasma CLZ levels are more likely to increase at the same doses than those of other ethnic groups [5,6]. Kikuchi et al. (2024) recently proposed that slow titration of CLZ reduces adverse inflammatory effects in Japanese patients [7].

Factors that inhibit CLZ metabolism include CYP1A2 inhibitors such as amiodarone, fluvoxamine, ciprofloxacin, oral contraceptives, and high doses of caffeine [8]. Additionally, obesity and chronic inflammation interfere with CLZ metabolism [8]. The role of CYP3A4 in CLZ metabolism is minor compared with that of CYP1A2; nevertheless, CYP3A4 and CYP3A5 play substantial roles [9,10,11,12,13]. Recently, Watanabe et al. (2024) reported that the concomitant use of lemborexant, which is a CYP3A4 inhibitor, increased CLZ plasma levels and approximately doubled the concentration-to-dose (C/D) ratio in one patient [14]. We had reported an increase in plasma CLZ levels with concomitant suvorexant treatment in a patient with schizophrenia [15]. These reports suggest that lemborexant and suvorexant, which are CYP3A4 inhibitors, increase CLZ plasma levels. Both Watanabe et al. (2024) and our group observed an approximate doubling of the CLZ plasma level, which should alert clinicians to the considerable risk of adverse side effects due to the rapid and extreme increase in CLZ concentration. 

Moreover, various pharmacokinetic factors, such as single nucleotide polymorphisms (SNPs) of drug-metabolizing enzymes and transporters, have been implicated in dynamic changes in CLZ levels. Increase in CLZ plasma C/D is associated with the drug efflux transporter *ABCG2* (421C>A) [16], CLZ-induced agranulocytosis, and liver organic anion transporting polypeptide (OATP, gene name: *solute carrier organic anion transporter*, (*SLCO*)) [17]. The relationship between OATP and granulocyte reduction has been identified through genome-wide association studies (GWAS), but its direct impact remains unclear.

OATPs are Na^+^-independent transporters expressed in various tissues (liver, kidney, and small intestine). They enable cellular membrane transport of endogenous substances such as bile acids and various drugs [18,19,20,21,22,23,24,25,26,27]. Hepatic transporters, such as OATP, regulate access to hepatocellular enzymes and are transported into the bile canaliculi. Hepatic uptake via transporters may be the rate-limiting step in the overall hepatic drug clearance process [28,29]. OATP1B1, OATP1B3, and OATP2B1 are localized to the sinusoidal membrane of hepatocytes and transport a wide variety of clinically used drugs and endogenous compounds [18,19,23,24,25,26,27,30,31].

Therefore, this clinical study investigated the relationship between SNPs of various pharmacokinetic factors (drug-metabolizing enzymes and transporters) and dynamic changes in CLZ levels. We aimed to determine whether CLZ acts as a substrate for pharmacokinetic factors using in vitro assays and molecular docking calculations.

## 2. Results

### 2.1. Quantitation of CLZ and N-Desmethyl CLZ (norCLZ) Plasma Concentrations in Patients with TRS

CLZ and norCLZ plasma concentrations were quantified as described previously [32]. The parameters are listed in Table 1. The raw data and CLZ and norCLZ plasma concentration calculations, norCLZ/CLZ ratio, and other CLZ exposure parameters are shown in Appendix A. The CLZ trough values of four patients (nos. 2, 4, 5, and 7) were higher than the upper limit of the therapeutic range (350–600 ng/mL), and those of three patients (nos. 1, 6, and 9) were lower than the lower limit. The C/D ratios of three patients (nos. 2, 4, and 7) were relatively high (more than 2/L), and the C/D/kg ratio of two patients (nos. 3 and 7) was relatively high (more than 0.050/L·kg) among the patients. CLZ metabolic efficiency (norCLZ/CLZ ratio) was calculated by quantifying norCLZ. The norCLZ/CLZ ratio of four patients (nos. 4, 6, 8, and 10) was higher than the upper limit, and that of one patient (no. 2) was lower than the lower limit of the reported range.

### 2.2. Drug Metabolizing Enzyme Genotypes and CYP2D6 Copy Number in Patients with TRS

Major SNPs of drug-metabolizing enzymes were analyzed to clarify whether their activities affected the CLZ parameters (CLZ trough, norCLZ/CLZ ratio, or C/D/kg ratio). Alleles were classified according to their perceived functionality based on activity score (AS) assignments. Phenotype status was classified as poor metabolizer (PM), intermediate metabolizer (IM), or extensive metabolizer (EM) for CYP2C19 and CYP2D6. The results are summarized in Table 2. The numbers of patients with **1A/*1A*, **1A/*1F*, and **1F/*1F* CYP1A2 genotypes were two (nos. 7 and 10), five (nos. 1, 2, 5, 6, and 8), and three (nos. 3, 4, and 9), respectively. One patient (no. 2) was a PM, and four patients (nos. 1, 2, 6, and 10) were IMs of CYP2C19. Two patients (nos. 8 and 9) were IMs of CYP2D6. Three patients (nos. 4, 7, and 10) showed **1/*1G* CYP3A4, two patients (nos. 7 and 10) showed **1/*3* CYP3A5, and eight patients (nos. 1–6, 8, and 9) showed **3/*3* CYP3A5 mutations. One patient (no. 1) showed the homozygous mutation 769G>A of flavin-containing monooxygenase (FMO) 3 and three patients (nos. 4, 6, and 10) showed heterozygous mutations such as 472G>A, 769G>A, and 923A>G of FMO3. All patients harbored CYP2C9 WT. Subsequent PCR assessment showed that two patients (nos. 8 and 9) had one copy, and eight patients (nos. 1–7 and 10) had two copies of *CYP2D6* (Figure 1).

### 2.3. Drug Transporter Genotypes in Patients with TRS

Major SNPs (missense or deletion) of drug transporters were analyzed to clarify whether those activities affect CLZ parameters (CLZ trough, norCLZ/CLZ ratio, or C/D/kg ratio). The results are summarized in Table 3. Two patients (nos. 3 and 10) showed the homozygous mutation 2677G>T/A of P-glycoprotein (Pgp), and seven patients (nos. 1, 2, 4, and 6–9) showed heterozygous mutation. Two patients (nos. 6 and 7) showed homozygous mutation 421C>A of the breast cancer resistance protein (BCRP), and three patients (nos. 8–10) showed heterozygous mutation. The other patients harbored WT or 34G>A mutations, which did not induce BCRP expression or functional alteration. The number of patients with **1a/*1a*, **1a/*1b*, **1b/*1b*, **1b/*15*, and **15/*15* OATP1B1 was 1 (no. 3), 2 (nos. 1 and 8), 2 (nos. 5 and 10), 3 (nos. 2, 6, and 9), and 2 (nos. 4 and 7), respectively. Four patients (nos. 3, 5, 8, and 10) showed homozygous mutations 334T>G and 699G>A in OATP1B3, and three patients (nos. 1, 6, and 9) showed heterozygous mutations. Patients 3, 7, and 8 harbored multiple OATP1B3 mutations. The numbers of patients with **3* homo, **3* hetero, 935G>A homo, 935G>A hetero, 601G>A hetero, and 76_84del of OATP2B1 were 1 (no. 3), 4 (nos. 4, 7, 8, and 10), 1 (no. 3), 4 (nos. 2, 6, 7, and 8), 1 (no. 7), and 1 (no. 7), respectively. Only patient 1 showed a heterozygous mutation 913A>T of organic anion transporter (OAT) 3. All patients were of the OATP4C1 WT genotype.

### 2.4. CLZ Uptake in OATP1B1-, OATP1B3-, and OATP2B1-Expressing Human Embryonic Kidney (HEK293) Cells

As multiple OATP mutations were observed in patients with high CLZ parameters, CLZ uptake via OATPs was examined to clarify whether CLZ was a substrate. Uptake experiments performed within 15 min to assess OATP-mediated initial uptake, which reflects transporter activity, showed that CLZ uptake was higher in OATP1B1- and OATP1B3-expressing HEK293 cells than in mock-transfected cells (Figure 2A,B). A significant increase was observed at 1, 10, and 15 min for OATP1B1 and at 1 min for OATP1B3 (*p* < 0.05). Transporter-mediated CLZ uptake was observed more frequently in OATP1B1-expressing HEK293 cells than in mock cells. However, compared with that of the mock cells, CLZ uptake by OATP1B3- and OATP2B1-expressing HEK293 cells did not increase in a time-dependent manner (Figure 2B,C).

### 2.5. Concentration Dependence of OATP1B1-Mediated CLZ Uptake

We examined the initial rate of CLZ concentration-dependent transport via OATP1B1. OATP1B1-mediated CLZ uptake was quantified by subtracting the uptake by mock cells from the uptake by OATP1B1/HEK293 cells at each point. As shown in Figure 3, OATP1B1-mediated uptake was saturable at higher CLZ concentrations and was used to establish Michaelis–Menten kinetics. The *K*_m_ and *V*_max_ values for CLZ uptake via OATP1B1 were 38.9 ± 4.9 µM and 2752 ± 831 pmol/mg protein/10 min, respectively. The *K*_m_ value for CLZ uptake was within the previously reported range [33].

### 2.6. Inhibitory Effects of Typical Inhibitors for Drug Transporters

Evaluating the effects of transporter inhibitors on OATP (negative control: *p*-aminohippuric acid (PAH), which is an OAT inhibitor) showed that cyclosporin A and rifampicin, which are strong OATP inhibitors, inhibited OATP1B1-mediated uptake by 37.5% and 20.6%, respectively. However, these effects were relatively weak and not statistically significant (Figure 4). Other substrates or OATP inhibitors did not significantly inhibit OATP1B1-mediated CLZ uptake (Figure 4).

### 2.7. Predicted CLZ Binding Structures on OATP1B1

Molecular docking program-based prediction of binding positions of CLZ and cyclosporin A on OAP1B1 showed that OATP1B1 exists in two states as follows: outward-open and inward-open; the structures of both states have been determined using cryogenic electron microscopy (Cryo-EM) recently [34,35]. We modeled the binding structure of the ligands for both the outward- and inward-open states (Figure 5); these binding poses shared sites with the binding poses of known substrates. Shan et al. (2023) classified the OATP1B1 binding pockets in an outward-open state into a “major” and a “minor” pocket (Appendix A) [34]. The predicted CLZ binding site corresponded to the “minor” pocket, whereas that of cyclosporin corresponded to the “major” pocket. Several candidate positions were generated using a docking program. The poses ranked first are shown in Figure 5. In the case of CLZ in the outward-open state, the first- and second-ranked poses were located in a shallow “minor” pocket, whereas the third-ranked pose was located in a deep “minor” pocket (Appendix A). The third-ranked pose was closer to the experimentally determined pose of another compound (2’,7’-dichlorofluorescein; Appendix A) and has been selected for display in Figure 5. We modeled the binding positions of cyclosporin A using a similar procedure. Notably, cyclosporin A is a flexible molecule with a high degree of freedom, which makes exploring its conformations challenging. The predicted binding poses of cyclosporin were located in the “major” pocket; those of cyclosporin A and CLZ did not significantly overlap in the outward-open state (Figure 5A and Appendix A). Similar docking calculations were performed for the inward-open OATP1B1 structures (Figure 5C,D). The predicted CLZ binding site in the inward-open state was similar to the experimental binding site for estrone 3-sulfate (Appendix A). The predicted binding sites of CLZ and cyclosporine A in the inward-open state were located differently but overlapped more frequently than those in the outward-open state. Figure 5B,D show details of the predicted CLZ binding sites. The amino acids of OATP1B1 at the four predicted binding sites (Y352, A355, F386, and L545) are shown in Figure 5B,D. When the alignment was generated using the ClustalW2 program [36], these were different from those of OATP1B3 and OATP2B1 (Appendix A).

## 3. Discussion

In this study, we explored the relationship between SNPs of pharmacokinetic factors, including metabolizing enzymes and transporters, and dynamic changes in CLZ plasma levels. Furthermore, we evaluated whether CLZ acts as a substrate for these pharmacokinetic factors using in vitro assays and molecular docking analyses. We found that CLZ is a substrate of OATP1B1, and the potential alteration of CLZ pharmacokinetic parameters is because of the presence of SNPs in OATPs.

CLZ trough values help determine whether they were within the therapeutic range. We found that the CLZ trough values were higher than the therapeutic window (350–600 ng/mL) for four patients and lower for three. Patient no. 5 was prescribed metoclopramide, which is a reversible CYP2D6 inhibitor [37] that possibly reduces CLZ metabolism and elevates the CLZ trough value. Although CYP2D6 was one of the CYPs that contribute to CLZ metabolism [38,39], the clinical impact of CYP2D6 on CLZ pharmacokinetics should be clarified before confirming the involvement of metoclopramide. The C/D ratio was relatively high (>2 ng/mL/mg/day) among three patients. The C/D ratio is reportedly higher in individuals of Asian ancestry than in individuals of European ancestry [40]. The average C/D ratio was 1.6 (Appendix A), which was similar to the reported value of 1.8 [41]. The C/D/kg ratio of the two patients was relatively high (>0.050 /L·kg) among the patients. Patient 7 seemed to be exposed to relatively high CLZ concentrations compared with those of the other patients, as confirmed by the CLZ trough value, C/D ratio, and C/D/kg value. None of the patients with high CLZ trough values, C/D ratios, or C/D/kg values had liver or renal failure, which could affect CLZ pharmacokinetics.

The metabolic efficiency of CLZ (norCLZ/CLZ ratio) was calculated by measuring norCLZ levels. In total, four and one patient showed a higher or lower norCLZ/CLZ ratio than that of the therapeutic window (0.45–0.79), respectively. Two (nos. 6 and 10) among the four patients with a high norCLZ/CLZ ratio were obese (body mass index (BMI) of ≧25). However, obesity could not explain the high norCLZ/CLZ ratio because CLZ suppresses metabolic activity (the norCLZ/CLZ ratio should be low) and contributes to the increase in CLZ plasma concentration [42]. Patient 10 was an active smoker among the four patients with a high norCLZ/CLZ ratio, and the smoking status may have contributed to the elevated norCLZ/CLZ ratio because of CYP1A2 induction.

We investigated the relationship between SNP of various pharmacokinetic factors and CLZ dynamic change in CLZ trough value, the norCLZ/CLZ ratio, the C/D ratio, and the C/D/kg value. Major SNPs of drug-metabolizing enzymes were analyzed to clarify whether these activities affected CLZ parameters. In total, 2, 5, and 3 patients, respectively, showed **1A/*1A*, **1A/*1F*, and **1F/*1F* CYP1A2 mutations. CYP1A2 **1F/*1F* in intron 1 was suggested to confer higher inducibility of CYP1A2 through smoking, with a 1.6-fold higher metabolic activity in smokers than in smokers with other genotypes, whereas no differences were found between genotypes in nonsmokers [43,44]. Only patient 10 was a smoker, and patients 3, 4, and 9 with CYP1A2 **1F/*1F* did not smoke. Thus, CYP1A2 **1F/*1F* alone cannot explain the alterations in CLZ parameters. Patient 2 was a PM of CYP2C19, and patients 1, 2, 6, and 10 were IMs. Patients 8 and 9 were IM of CYP2D6. Although CYP2C19 and CYP2D6 are reportedly involved in CLZ metabolism, the clinical significance of these molecules requires further consideration [45]. Patients 4, 7, and 10 showed **1/*1G* CYP3A4 mutation. Significantly lower dose-adjusted C_0_ values of tacrolimus in the patients with the CYP3A4**1G* allele compared with those with the CYP3A4**1/*1* genotype were reported [46]. Two (nos. 7 and 10) and eight (nos. 1–6, 8, and 9) patients showed **1/*3* and **3/*3* CYP3A5 genotypes, respectively. CYP3A5**3* reportedly caused alternative splicing and protein truncation, which resulted in the absence of CYP3A5 in the tissues of some individuals [47]. Allele *3 was reportedly more frequent (74.0%) than that of other variant alleles in Japanese patients, and the genotype frequencies of CYP3A5**1/*1*, **1/*3*, and **3/*3* were 7.9%, 35.5%, and 55.9%, respectively [48]. If CLZ is the substrate of CYP3A5, the plasma concentrations of patients 7 and 10 with **1/*3* should be lower than those of other patients. However, these alterations were not observed in the patients analyzed in this study. Homozygous mutation (769G>A) of FMO3 was observed in patient 1, and heterozygous mutations such as 472G>A, 769G>A, and 923A>G of FMO3 were observed in patients 4, 6, and 10. FMO3 472G>A (Glu158Lys) and 923A>G (Glu308Gly) were previously identified in a Japanese female proband with trimethylaminuria and were reportedly inherited from her mother [49]. All variants of FMO3, including 769G>A (Val257Met), are involved in the reduction of FMO3 activity with respect to the *N*-oxygenation of trimethylamine [49]. However, these SNPs did not explain the low or high CLZ values.

Major SNPs (missense or deletion) of drug transporters were analyzed to clarify whether these activities affected CLZ parameters. The results are summarized in Table 3. Patients 3 and 10 showed homozygous mutation 2677G>T/A in Pgp, and patients 1, 2, 4, and 6–9 showed heterozygous mutation. Patients 6 and 7 showed homozygous mutation 421C>A of BCRP, and patients 8–10 showed heterozygous mutation. These variants induce expressional or functional alterations in Pgp and BCRP, which cause substrate accumulation for these transporters. Previously, the 2677G > T polymorphism was reported to have no significant influence on CLZ, norCLZ, or CLZ+norCLZ plasma concentrations [50]. The involvement of BCRP 421C>A in a higher CLZ C/D ratio has been reported [16]. In our study, CLZ accumulation or a high norCLZ/CLZ ratio was observed not only in patients with Pgp 2677G>T/A or BCRP 421C>A but also in patients with heterozygous mutations or WT of those transporters. Thus, these transporters alone could not explain the alterations in CLZ parameters.

Genetically, OATP1B1**1b* (388A>G (rs2306283; Asn130Asp)) and **5* (521T>C (rs4149056; Val174Ala)) are well-known single-nucleotide variants with amino acid substitutions in OATP1B1. Among the four haplotypes formed by single nucleotide variations of **1b* and **5* (OATP1B1**1a* (WT), OATP1B1**1b*, OATP1B1**5*, and OATP1B1**15* (both **1b*&**5*)) [51], the common variant OATP1B1**5* was highlighted using a GWAS, suggesting an increased risk of simvastatin-induced myopathy in variant carriers [52]. The reduced hepatic uptake of OATP1B1 substrates was supported by in vitro experiments using cell lines that stably express OATP1B1**5* or **15* variants [53,54]. The pharmacokinetics of mycophenolic acid in stable renal transplant patients on a cyclosporin-free regimen are influenced by genetic polymorphisms (334T>G and 699G>A) of OATP1B3, of which mycophenolic acid glucuronide is a substrate [55]. Polymorphisms of OATP2B1**3* (1457C>T (rs2306168; Ser486Phe)) are genetic factors that affect OATP2B1 functions [33]. OATP2B1**3* was associated with reduced transport of substrates by OATP2B1 [56].

In our study, the number of patients with **1a/*1a*, **1a/*1b*, **1b/*1b*, **1b/*15*, and **15/*15* of OATP1B1 was one (no. 3), two (nos. 1 and 8), two (nos. 5 and 10), three (nos. 2, 6, and 9), and two (nos. 4 and 7), respectively. Homozygous mutations 334T>G and 699G>A in OATP1B3 were observed in four patients (nos. 3, 5, 8, and 10), and heterozygous mutations were observed in three patients (nos. 1, 6, and 9). Patients 3, 7, and 8 harbored multiple *SLCO1B3* (OATP1B3) mutations. The numbers of patients with **3* homo, **3* hetero, 935G>A homo, 935G>A hetero, 601G>A hetero, and 76_84del of OATP2B1 were one (no. 3), four (nos. 4, 7, 8, and 10), one (no. 3), four (nos. 2, 6, 7, and 8), one (no. 7), and one (no. 7), respectively. Although multivariate analysis of OATPs variants and CLZ parameters could not be performed owing to the small sample numbers, patients with multiple OATP variants seemed to be highly exposed to CLZ and/or norCLZ (nos. 4, 5, 6, 7, 8, and 10).

The heterozygous mutation 913A>T in OAT was observed in patient 1 only. All patients had OATP4C1 WT genotype. Thus, OAT3 and OATP4C1 may not be involved in the alteration of CLZ parameters.

Additionally, we performed a CLZ uptake study using OATP-expressing HEK293 cells to determine whether CLZ is a substrate for OATP1B1, OATP1B3, and OATP2B1. CLZ uptake by OATP1B1-expressing HEK293 cells increased significantly. Additionally, we examined the concentration dependence of OATP1B1-mediated CLZ uptake. The uptake was saturable at higher CLZ concentrations, and the uptake approximated Michaelis–Menten kinetics. The *K*_m_ and *V*_max_ values of CLZ uptake via OATP1B1 were 38.9 µM and 2752 pmol/mg protein/10 min, respectively. The *K*_m_ value for CLZ uptake was within the range reported previously [33]. We also evaluated the effects of transporter inhibitors on OATP (negative control: PAH, which is an OAT inhibitor). The result revealed that the strong inhibitors for OATP (cyclosporin A and rifampicin) showed weak or moderate inhibitory effects on OATP1B1-mediated CLZ uptake (Figure 4). Therefore, certain OATP1B1 inhibitors show discrepancies in their inhibitory potential depending on the substrate used in the cell-based assay [57]. This may be attributed to the differences in the binding sites of the substrates and inhibitors, which have been reported for other OATP molecules [58]. However, further studies are required to elucidate the mechanisms underlying this phenomenon.

Molecular docking calculations helped investigate the interactions between CLZ and OATPs in detail. The docking program identified the likely binding poses of CLZ in both the outward-open and inward-open states, which were similar to known binding poses of other substrates. Additionally, the predicted binding structures of CLZ and cyclosporin A did not overlap in the outward-open or inward-open states of OATP1B1 (Figure 5, Appendix A). This may explain the weak inhibitory effect of cyclosporine A on CLZ uptake via OATP1B1 (Figure 4). Additionally, we found that the amino acids of OATP1B1 in the four predicted binding sites (Y352, A355, F386, and L545) shown in Figure 5B,D were different from those of OATP1B3 and OATP2B1 (Appendix A). The difference in the amino acids of the predicted binding sites between the OATPs may explain the difference in OATP-mediated CLZ uptake, as shown in Figure 2.

In conclusion, we investigated the relationship between major SNPs of drug-metabolizing enzymes, transporters, and CLZ parameters. Our findings demonstrate that the SNPs of OATPs are involved in the alteration of CLZ parameters. Direct examination using in vitro uptake study showed that CLZ is an OATP1B1 substrate; molecular docking calculations supported the uptake results. This is the first study to show that CLZ is a substrate of OATP1B1, and the potential alteration of CLZ pharmacokinetic parameters is because of the presence of SNPs in OATPs. We believe that this study provides novel information for clinical staff and researchers in the pharmaceutical industry.

### Limitations of This Study

One of the key limitations of this study is the absence of an analysis linking clozapine plasma concentrations to clinical improvements in schizophrenia symptoms. While the therapeutic effectiveness of clozapine is a critical consideration in clinical practice, the primary aim of this study was to investigate the pharmacokinetic variability and its potential genetic and pharmacological determinants, rather than evaluating clinical outcomes. We did not include measures of symptom improvement or therapeutic response due to the limited sample size, the retrospective nature of this study, and the complexity involved in assessing schizophrenia outcomes across a heterogeneous patient population. Additionally, although we carefully analyzed the available data, no consistent or discernible pattern in absolute plasma concentrations, dosage-normalized concentrations, or norCLZ/CLZ ratios could be identified. This variability likely reflects the multifactorial nature of clozapine pharmacokinetics, influenced by a complex interplay of genetic, metabolic, and environmental factors. Future studies with larger sample sizes, prospective designs, and integrated clinical outcome measures will be essential to further elucidate the relationship between clozapine plasma concentrations, pharmacokinetic variability, and therapeutic efficacy. Such research could provide more actionable insights for optimizing clozapine therapy in treatment-resistant schizophrenia.

## 4. Materials and Methods

### 4.1. Ethics

This study was conducted in accordance with the Declaration of Helsinki and approved by the Ethics Committee (project numbers: 2023-1-618-1 (date of approval: 28 October 2019) and 2024-1-123 (date of approval: 28 June 2021)). Written informed consent was obtained from each participant prior to this study.

### 4.2. Patients

Ten patients were enrolled in this study. The patient characteristics are listed in Table 4. The mean (range) age and BMI were 39 years (range, 26–50) and 20.7 kg/m^2^ (range, 16.7–29.1), respectively. Parameters such as obesity and smoking status, which affect CLZ plasma concentration, are also listed.

### 4.3. Drug Treatment

All patients received CLZ (Clozaril; Novartis Pharma Ltd., Tokyo, Japan) and no other psychotropic drugs. Their adherence was confirmed by medical doctors, nursing staff, or their families. Blood samples were collected immediately before drug administration (through sampling). CLZ and norCLZ plasma levels were quantified according to an established procedure [32].

### 4.4. Genotyping of Drug Metabolizing Enzymes and Transporters Gene Polymorphisms

Genomic DNA was extracted from blood samples using a standard procedure. Primer pairs for amplification of promoters, introns, and exons of target genes are listed in Appendix A. Genomic DNA (10–20 ng) was amplified using 2× AmpliTaq Gold™ 360 Master Mix (Thermo Fisher Scientific Inc., Waltham, MA, USA). The reaction mixture contained 10–20 ng of genomic DNA, 0.5 mM of each primer, and 2× AmpliTaq Gold™ 360 Master Mix (final volume, 20 mL). PCR thermal profiles consisted of an initial denaturation at 95 °C for 10 min, followed by 30 repetitive cycles of 95 °C for 30 s, annealing at 60 °C, and extension at 72 °C for 1 min; final extension was at 72 °C for 7 min.

Genotyping of CYP2D6 polymorphisms and copy number assays were performed as previously reported [59]. For whole *CYP2D6* amplification, genomic DNA (10–20 ng) was amplified using 50x PrimeSTAR^®^ GXL DNA Polymerase (TaKaRa, Shiga, Japan). The reaction mixture contained 10–20 ng of genomic DNA, 0.2 mM of each primer, and 50× PrimeSTAR^®^ GXL DNA Polymerase (final volume, 20 mL). The PCR thermal profile consisted of initial denaturation at 95 °C for 5 min, followed by 30 cycles of 98 °C for 10 s, annealing and extension at 68 °C for 5 min, and a final extension at 72 °C for 7 min. CYP2D6 exon amplification was performed using the same PCR enzyme, reaction mixture composition, and PCR thermal profiles. Control DNA samples containing two copies were used.

### 4.5. Materials

CLZ was purchased from Santa Cruz Biotechnology (Santa Cruz, CA, USA), and norCLZ was purchased from Sigma-Aldrich (St. Louis, MO, USA). The plasmid vector (pRP[Exp]-Neo-CAG>hSLCO2B1[NM_007256.5]) was purchased from VectorBuilder (Kanagawa, Japan), and it was used to establish the human OATP2B1 cell line. All chemicals used were of the highest commercially available purity.

### 4.6. Cell Culture

HEK293 cells were transduced with OATP1B1, OATP1B3, or an empty vector, as previously described [60]. We established a human OATP2B1-stably expressing cell system using the purchased vector and HEK293 cells. After selection with G418, single colonies expressing OATP2B1 were screened using a typical substrate (fexofenadine) for transport studies (Appendix A). OATP1B1/HEK293, OATP1B3/HEK293, OATP2B1/HEK293, and mock cells were maintained in Dulbecco’s modified Eagle’s medium supplemented with 10% fetal bovine serum (Gibco, Thermo Fisher Scientific Inc.) and G418 (0.5 mg/mL; Nacalai Tesque, Inc., Kyoto, Japan) at 37 °C under 5% CO_2_ and 95% humidified air.

### 4.7. Transport Study

CLZ cellular uptake was measured in a cell monolayer cultured in 24-well plates. Cells were seeded at a density of 4.0 × 10^5^ cells/well and incubated in culture medium containing 5 mM sodium butyrate for 24 h before the uptake study. The cells were washed once with Krebs–Henseleit (KH) buffer (118 mM NaCl, 23.8 mM NaHCO_3_, 4.83 mM KCl, 0.96 mM KH_2_PO_4_, 1.20 mM MgSO_4_, 12.5 mM *N*-(2-hydroxyethyl) piperazine-*N’*-2-ethanesulfonic acid, 5.0 mM D-glucose, and 1.53 mM CaCl_2_; pH 7.4), followed by preincubation in KH buffer. Cellular uptake was initiated by adding CLZ-containing KH buffer with or without transporter inhibitors. Uptake was terminated after the indicated times by replacing the incubation buffer with ice-cold KH buffer. After washing the cells twice with ice-cold KH buffer, CLZ concentration was measured using liquid chromatography/tandem mass spectrometry [32]. Cellular uptake was presented as the uptake quantity divided by the amount of cellular protein quantified using the Bradford protein assay. The experiment was performed in triplicate (*n* = 3) and repeated thrice. The compounds were dissolved in dimethyl sulfoxide (DMSO) to a final DMSO concentration of <0.5%.

### 4.8. Molecular Docking Calculations

The program AutoDock Vina 1.2.5 was used to perform molecular docking [61,62] of CLZ and cyclosporin A on two different states of OATP1B1 [34,35]. We followed an ensemble docking approach by preparing multiple conformations for both the ligand and receptor [63,64] and performing molecular docking calculations for all possible combinations of the conformations using the *fkcombu* [65] and *ghecom* [66] programs (Appendix A). All binding ligand poses generated from the various ligand–receptor conformation pairs and random seeds were clustered using the single-linkage clustering method with a threshold root mean square deviation (RMSD) of 2.0 Å for ligand atoms. The details of the modeling are described in the Appendix A.

### 4.9. Statistical Analysis

Data are expressed as mean ± standard error (S.E.). Differences between groups were tested for significance using the unpaired Student’s t-test as was appropriate. Multiple statistical comparisons were performed using one-way analysis of variance (ANOVA) followed by Tukey’s test. Statistical analyses were performed using JMP Pro 17 software (ver. 17.2.0, SAS Institute Inc., Cary, NC, USA). Statistical significance was set at *p* < 0.05.

## Figures and Tables

**Figure 1 ijms-25-13228-f001:**
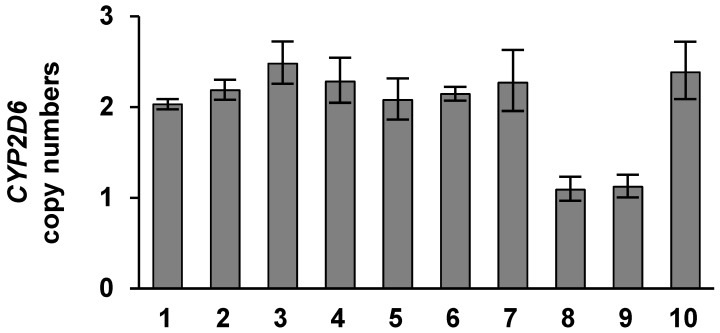
CYP2D6 copy number assay. The X-axis represents the patient numbers. The Y-axis represents the copy numbers of CYP2D6. Data are presented as the mean ± standard error (S.E.) (*n* = 3).

**Figure 2 ijms-25-13228-f002:**
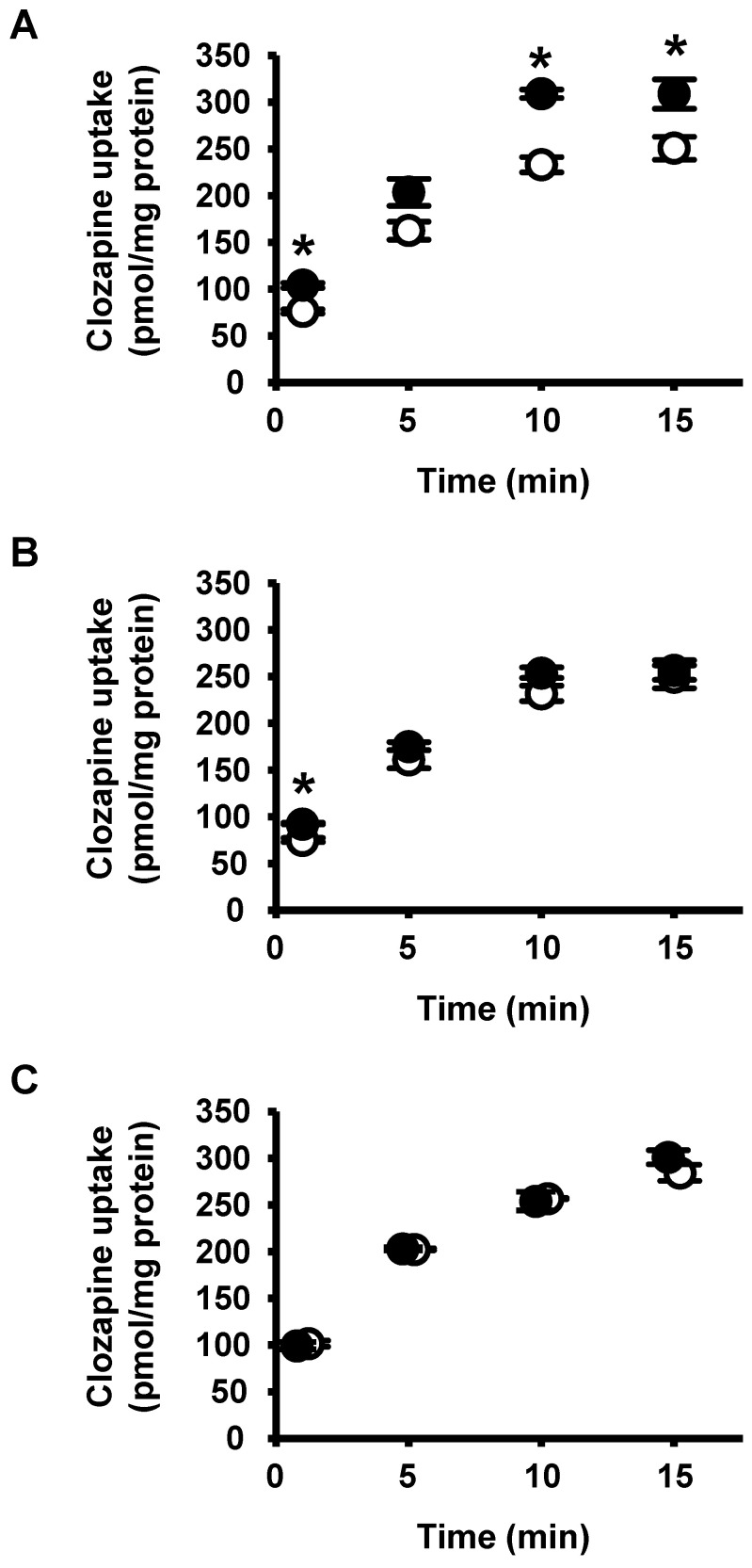
Time-dependent CLZ uptake by OATP1B1-, OATP1B3-, and OATP2B1-expressing HEK293 cells. Cells were incubated with 1 µM CLZ for 1, 5, 10, and 15 min at 37 °C in KH buffer (pH 7.4). (**A**) Open and closed circles represent mock cells (○) and OATP1B1-expressing HEK293 cells (●), respectively. An asterisk indicates a significant difference from the value of mock cells (*p* < 0.05). (**B**) Open and closed circles represent mock cells (○) and OATP1B3-expressing HEK293 cells (●), respectively. An asterisk indicates a significant difference from the value of mock cells (*p* < 0.05). (**C**) Open and closed circles represent mock cells (○) and OATP2B1-expressing HEK293 cells (●), respectively. Data are presented as the mean ± standard error (S.E.) (*n* = 3).

**Figure 3 ijms-25-13228-f003:**
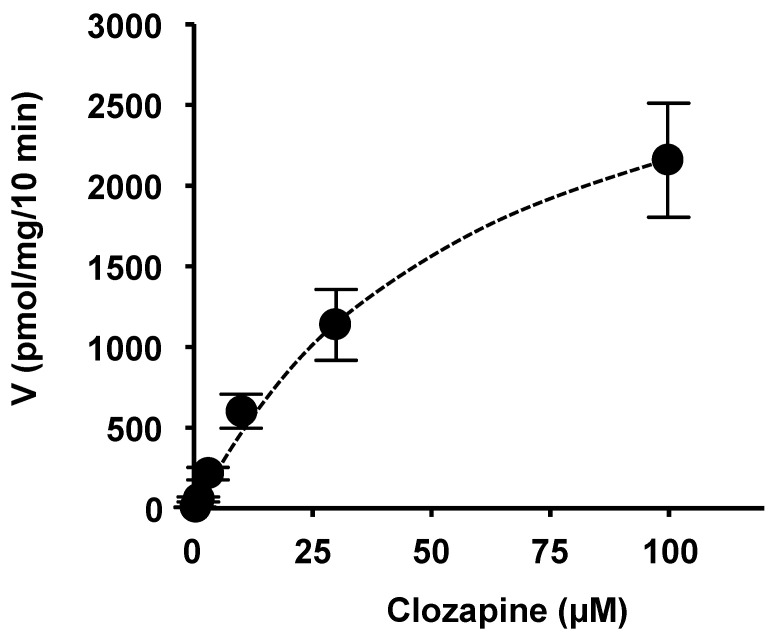
Concentration dependence of OATP1B1-mediated CLZ uptake. Cells were incubated with 0.3, 1, 3, 10, 30, and 100 µM of CLZ for 10 min at 37 °C. The quantity of OATP1B1-mediated uptake was calculated by subtracting the nonspecific uptake of CLZ by mock cells from the total cellular uptake by OATP1B1-expressing cells. Data are shown as mean ± S.E. (*n* = 6).

**Figure 4 ijms-25-13228-f004:**
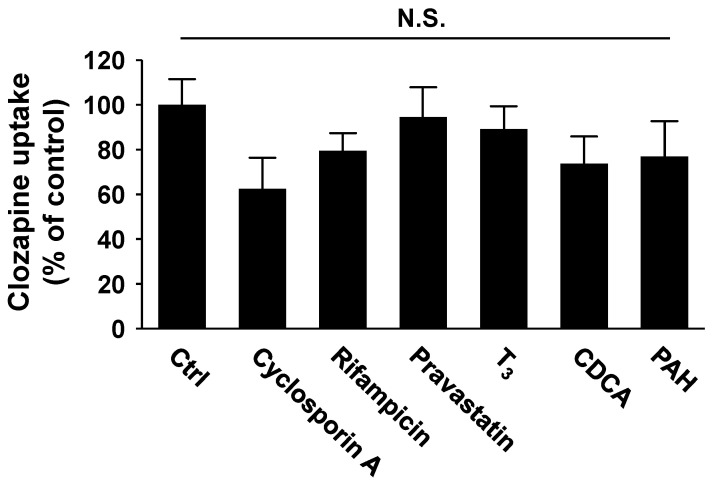
Inhibitory effects of typical inhibitors for drug transporters on OATP1B1-mediated CLZ uptake. Cells were incubated with 1 µM CLZ and the inhibitors for 10 min at 37 °C. The concentrations of cyclosporin A, rifampicin, pravastatin, triiodothyronine (T_3_), chenodeoxycholic acid (CDCA), and *p*-aminohippuric acid (PAH) were 10, 10, 50, 50, 10 µM, and 1 mM, respectively. The quantity of OATP1B1-mediated uptake was calculated by subtracting the nonspecific uptake of CLZ by mock cells from the total cellular uptake by OATP1B1-expressing cells. Data are shown as mean ± S.E. (*n* = 3). Multiple statistical comparisons were made using one-way analysis of variance (ANOVA), followed by Tukey’s test. (N.S., no significant difference).

**Figure 5 ijms-25-13228-f005:**
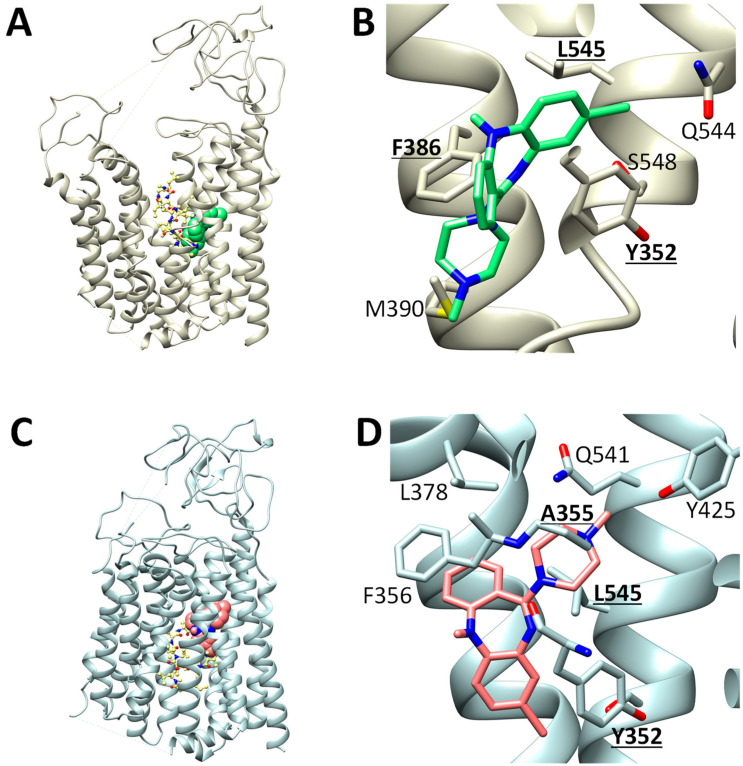
Predicted binding poses of CLZ and cyclosporin A with OATP1B1. (**A**) Predicted binding poses of CLZ (green spacefill model) and cyclosporin A (yellow ball-and-stick model) with Cryo-EM structure of OATP1B1 in outward-open state (PDB ID: 8k6l). (**B**) Predicted binding pose of CLZ on OATP1B1 in outward-open state. Representative conformation of the 3rd rank cluster is shown. An amino acid of OATP1B1 in three predicted binding sites (Y352, A355, and L545) was different from those of OATP1B3 and OATP2B1 (Appendix A). (**C**) Cryo-EM structure of OATP1B1 in inward-open state (PDB ID: 8hnd) with predicted binding poses of CLZ (red spacefill model) and cyclosporin A (yellow ball-and-stick model). (**D**) Predicted binding pose of CLZ on OATP1B1 in an inward-open state. Representative conformation of the 1st rank cluster is shown. An amino acid of OATP1B1 in three predicted binding sites (Y352, F386, and L545) was different from those of OATP1B3 and OATP2B1 (Appendix A).

**Table 1 ijms-25-13228-t001:** Plasma concentrations of CLZ and norCLZ and other parameters of CLZ exposure and metabolism in 10 patients.

Patient No.	CLZ (ng/mL) (C)	C/Dose (/L)	C/Dose/kg (/L·kg)	norCLZ (ng/mL)	norCLZ/CLZ
1	248–346	1.2–1.7	0.021–0.029	158–220	0.58–0.70
2	395–672	1.3–2.2	0.028–0.046	138–319	0.35–0.48
3	347	1.3	0.064	174	0.50
4	683	2	0.029	551	0.81
5	472–639	1.0–1.4	0.024–0.033	242–342	0.51–0.60
6	336–570	0.84–1.9	0.013–0.027	303–449	0.79–1.0
7	657–803	4	0.071	434–448	0.54–0.68
8	383	1.2	ND	432	1.13
9	291	0.7	0.013	257	0.67
10	383	1	0.011	433	1.13

**Table 2 ijms-25-13228-t002:** List of CLZ parameters and major genomic type of drug metabolic enzymes in 10 patients.

Patient No.	CLZ Trough	norCLZ /CLZ	C/Dose /kg	CYP1A2	CYP2C9	CYP2C19 (Phenotype)		CYP2D6 (Phenotype)		CYP3A4	CYP3A5	FMO3
1	Low			**1A/*1F*	WT	**1/*3*	(IM)	**10/*39*	(EM)	WT	**3/*3*	769G>A homo
2	High	Low		**1A/*1F*	WT	**2/*38*	(IM)	**1A/*1A*	(EM)	WT	**3/*3*	WT
3			High	**1F/*1F*	WT	**2/*2*	(PM)	**2/*10*	(EM)	WT	**3/*3*	WT
4	High	High		**1F/*1F*	WT	**1/*1*	(EM)	**1A/*10*	(EM)	**1/*1G*	**3/*3*	472G>A, 923A>G hetero
5	High			**1A/*1F*	WT	**1/*1*	(EM)	**1A/*10*	(EM)	WT	**3/*3*	WT
6	Low	High		**1A/*1F*	WT	**1/*2*	(IM)	**1A/*2*	(EM)	WT	**3/*3*	769G>A hetero
7	High		High	**1A/*1A*	WT	**1/*1*	(EM)	**1A/*2*	(EM)	**1/*1G*	**1/*3*	WT
8		High		**1A/*1F*	WT	**1/*1*	(EM)	**5/*10*	(IM)	WT	**3/*3*	WT
9	Low			**1F/*1F*	WT	**1/*1*	(EM)	**5/*10*	(IM)	WT	**3/*3*	WT
10		High		**1A/*1A*	WT	**2/*38*	(IM)	**1/*10*	(EM)	**1/*1G*	**1/*3*	769G>A hetero

CLZ: low, <350; high, >600 (therapeutic window, 350–600 ng/mL). norCLZ/CLZ: low, <0.45; high, >0.79 (therapeutic window, 0.45–0.79). C/Dose/kg: high, >0.05. EM, extensive metabolizer; IM, intermediate; PM, poor metabolizer.

**Table 3 ijms-25-13228-t003:** List of CLZ parameters and major genomic type of drug transporters in 10 patients.

Patient No.	CLZ Trough	norCLZ /CLZ	C/Dose /kg	Pgp	BCRP	OATP1B1	OATP1B3	OATP2B1	OAT3	OATP4C1
1	Low			2677G>T hetero	34G>A hetero	**1a/*1b*	334T>G, 699G>A hetero	WT	913A>T hetero	WT
2	High	Low		2677G>T hetero	WT	**1b/*15*	WT	935G>A hetero	WT	WT
3			High	2677G>T/A homo	34G>A hetero	**1a/*1a*	334T>G, 699G>A homo	935G>A, **3* homo	WT	WT
4	High	High		2677G>T hetero	WT	**15/*15*	WT	**3* hetero	WT	WT
5	High			WT	34G>A hetero	**1b/*1b*	334T>G, 699G>A homo	WT	WT	WT
6	Low	High		2677G>A hetero	421C>A homo	**1b/*15*	334T>G, 699G>A hetero	935G>A hetero	WT	WT
7	High		High	2677G>T hetero	421C>A homo	**15/*15*	WT	76_84del, 601G>A, 935G>A, **3* hetero	WT	WT
8		High		2677G>T hetero	421C>A hetero	**1a/*1b*	334T>G, 699G>A homo	935G>A, **3* hetero	WT	WT
9	Low			2677G>T hetero	34G>A, 421C>A hetero	**1b/*15*	334T>G, 699G>A hetero	WT	WT	WT
10		High		2677G>T/A homo	421C>A hetero	**1b/*1b*	334T>G, 699G>A homo	**3* hetero	WT	WT

CLZ: low, <350; high, >600 (therapeutic window, 350–600 ng/mL). norCLZ/CLZ: low, <0.45; high, >0.79 (therapeutic window, 0.45–0.79). C/Dose/kg: high, >0.05.

**Table 4 ijms-25-13228-t004:** Patients profile.

Number of patients (n)	10
Gender (n)	
Male	3
Female	7
Age (year, mean (range))	39 (26–50)
BMI (kg/m^2^, mean (range))	20.7 (16.7–29.1)
Obese/non-obese (n)	2/8
Smoking status (n)	
Still smoking	1
Quit smoking	1
Never smoked	7
Unknown	1
CLZ dose (mg/day, mean (range))	300 (175–500)

BMI, body mass index.

## Data Availability

Data are contained within the article or Appendix A.

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
