# Peer review of "Effect of Organic Anion Transporting Polypeptide 1B1 on Plasma Concentration Dynamics of Clozapine in Patients with Treatment-Resistant Schizophrenia"

_ijms, 2024, doi:10.3390/ijms252313228_

Round 1

Reviewer 1 Report

Comments and Suggestions for Authors

The authors have explored the effects of several new factors on pharmacodynamic of clozapine in schizophrenia patients and in vitro model. The main findings of this study are that clozapine is a substrate of organic anion transporting-polypeptide 1B1 (OATP1B1) and the presence of SNPs of this transporter could influence clozapine pharmacokinetic parameters. The authors have also analyzed influence of major genomic type of drug metabolic enzymes and drug transporter on clozapine concentration and its metabolism transformation to N-desmethyl clozapine showing that these factors influence clozapine pharmacodynamics. Moreover, uptake of clozapine by different variants of OATP as well as Km and Vmax values for uptake by OATPB1B were determined.

I have a minor suggestion to author to try to improve formatting of the figures as letters may be too large. Moreover, the appropriate year after citation should be added on several places, for example, line 69 Watanabe et al. should be changed to Watanabe et al. (2024). Finally, in journal IJMS the methodology section is the last one, and after introducing this change abbreviations should be carefully re-checked.

Reviewer 2 Report

Comments and Suggestions for Authors

GENERAL COMMENTS

The authors are commended for providing a large body of experimental findings. However, the major shortcoming of this manuscript is the failure of the authors to state which of the 10 investigated patients showed an improvement of their schizophrenia and how this improvement was measured. As stated in tables 2 (numbers) and 3 (qualitative description), the clozapine concentrations of these patients were all over the orienting therapeutic range, and neither absolute nor dosage-normalized concentrations norCLZ/CLZ ratios seem to follow an easily discernable pattern. The authors should help the reader finding such a pattern or clearly state that there was no such pattern.

SPECIFIC ITEMS

line 339, Discussion: The Discussion section starts rather abruptly. A short summary of the findings will help the reader.

Round 2

Reviewer 2 Report

Comments and Suggestions for Authors

Why did you delete the whole materials and methods section? With respect to the patient plasma samples investigated, this reviewer thinks that you still MUST declare the ethics committee number and vote.

Author Response

Thank you very much for your feedback. We would like to clarify that the Materials and Methods section has not been deleted but has been relocated to follow the Discussion section, in accordance with the journal’s author guidelines. This adjustment was made during the revision process to comply with the required manuscript structure. We sincerely apologize for any confusion this may have caused.

Regarding your second point, we appreciate your observation about the need to declare the ethics committee number and approval. We confirm that this information is already included in the revised manuscript (Materials and Methods section), and we have ensured that all ethical requirements for research involving patient plasma samples are fully detailed. If there is any additional clarification needed, we would be happy to provide it.

Round 3

Reviewer 2 Report

Comments and Suggestions for Authors

acceptable